# OpenReview forum: "How Does Vision-Language Adaptation Impact the Safety of Vision Language Models?"
_ICLR.cc/2025/Conference — ICLR 2025 Poster_

### Official Review · Reviewer_2icJ · 2024-10-28

**Soundness:** 3
**Presentation:** 3
**Contribution:** 2
**Rating:** 5
**Confidence:** 5

**Summary:**

Vision-Language (VL) adaptation enables Large Language Models (LLMs) to handle multimodal tasks, transforming them into Large Vision-Language Models (LVLMs). However, this paper finds that the VL adaptation process often reduces safety fine-tuning benefits, even with safe training data. Techniques like supervised fine-tuning or reinforcement learning from human feedback can help but still compromise safety and helpfulness due to over-rejection. Analysis in this paper shows VL adaptation may weaken certain safety-related layers, as VL adaptation and safety tuning objectives diverge, making simultaneous application suboptimal. This paper proposes weight merging as an effective solution to reduce safety degradation while maintaining model helpfulness.

**Strengths:**

This paper focuses on safety problems with Vision-Language models introduced by the vison-language adaptation. Their insights may help build more reliable and secure LVLMs.
1.	This paper show evidence that vision language models, e.g., LLaMA-2 Chat 7B and Tulu-2 7B, produce harmful responses caused by VL adaptation.
2.	Analysis of internal model weights suggests that VL adaptation may impact certain safety- related layers.
3.	To tackle this problem, this paper proposed a simple yet effective solution by merging weights.

**Weaknesses:**

1. Although the authors analyze specific models, such as LLaMA-2 Chat and Tulu-2, it is unclear whether these conclusions apply to other language models, which may differ in training data or parameter sizes (13B, 34B, etc.). To strengthen the paper's persuasiveness, the authors should consider conducting experiments on a wider range of models, such as such as Qwen2-VL-2B, Qwen2-VL-7B, LLAVA-V1.6-13B，ShareGPT4V-13B, Qwen-VL 13B , Yi-VL-34B, Qwen2-VL-72B, etc., to ensure the results have broad applicability.

2. The authors should select additional vision encoders and other connectors (e.g., the MLP used in this paper) for experiments to demonstrate the generalizability of the experimental results across different vision-language model structures.

3. The core solution of this paper is weight merging, but this method has been previously proposed by others (e.g., Prateek Yadav, Derek Tam, Leshem Choshen, Colin A Raffel, and Mohit Bansal. Ties-merging: Resolving interference when merging models. Advances in Neural Information Processing Systems, 36, 2024.).

4. Although the effectiveness of weight merging has been validated on selected models (such as LLaMA-2 Chat and Tulu-2), the authors need to test it on other model architectures (e.g., language models of different sizes or training approaches) to determine whether the method is broadly applicable to other models.

5. The trade-off between safety and effectiveness has not been fully addressed. Although the paper points out the trade-off between safety and effectiveness, the proposed solution (such as weight merging) mainly balances the two rather than providing a way to completely avoid this trade-off.

6. The authors primarily draw conclusions about conflicts based on experimental data but do not provide sufficient theoretical or technical explanations for why it is difficult to balance the goals of safety tuning and multimodal tasks. It's better to give more detailed analysis of why the objectives of safety tuning and multimodal tasks might be fundamentally in tension. It will be much better to develop a (potential) mathematical frameworks for understanding this trade-off.

**Questions:**

1. In Section 3.1, how is the filtering quality of VL adaptation data ensured? Are there any experimental data to support this?

2. The evaluation metric in line 299 seems to contain a possible error in description?

---

> ### Author Response · Authors · 2024-11-19
> **Rebuttal by Authors (1)**
>
> Dear reviewer 2icJ, we deeply appreciate your constructive feedback. Here we address the weaknesses (W) and questions (Q) that you raised on our work.
>
>
> **Model architecture ablations (W1, W2)**
>
> We recognize the importance of your observation, and would like to highlight that our experiment already addresses several backbone models. In Table 4, we present the results of experiments conducted by replacing the base LLM with Vicuna. Similarly, in Table 5, we assess safety and multimodal helpfulness for other open-source models such as LLaVA-Next 8B, Phi-3-Vision-Instruct, and MiniCPM-V-2.6, alongside LLaVA-v1.5. These findings reveal persistent safety issues regardless of LLM variations and VLMs, emphasizing the prevalence of safety degradation.
>
> To further address your feedback, we conduct additional experiments involving VL adaptations using LLaMA-2-7B, LLaMA-2-Chat-13B, and Mistral-7B-Instruct-v0.3 as base LLMs. We also explored the impact of substituting the vision encoder from OpenAI-LargeCLIP-336px to OpenAI-LargeCLIP-224px, and investigated the effects of replacing the connector module from an MLP to a single linear layer. These experiments confirmed our initial trends, reinforcing the robustness of our conclusions. We value your feedback and, if allowed, would like to include these additional findings in the final version.
>
> |Base LLM|Text-only ASR ↓|Multimodal ASR ↓|Multimodal helpfulness ↑|
> |-------------|--------------------|---------------------|----------------|
> |LLaMA-2-Chat 7B (Ours)|54.4|96.2|64.3|
> |LLaMA-2-Chat 13B|50.1|94.4|66.3|
> |Mistral 7B Instruct v0.3|52.1|95.3|65.7|
>
> |Ablation|Text-only ASR ↓|Multimodal ASR ↓|Multimodal helpfulness ↑|
> |-------------|--------------------|---------------------|----------------|
> |Ours|54.4|96.2|64.3|
> |CLIP ablation|57.9|96.7|61.8|
> |Connector ablation|55.7|96.9|62.2|
>
> ---
>
> **Discussions about model weight merging (W3, W4)**
>
> While there are previous works that apply weight merging to non-multimodal models to achieve well-rounded performance across various tasks, including safety, most of them focus on improving performance in specific knowledge or reasoning tasks rather than prioritizing safety. To the best of our knowledge, there has been no prior work addressing multimodal safety in this context.
> What makes model weight merging particularly significant for multimodal models is the unique nature of these models. Unlike non-multimodal models, multimodal models process multiple modalities simultaneously (in this work, both images and text), which results in significantly higher training costs. We believe that applying weight merging in this context has the potential to be a much more efficient approach for multimodal models, given their higher computational demands.
>
> We also conducted ablation studies that involved applying different merging methods beyond linear merging and varying the combinations of models targeted for merging (see Table 3), as well as adjusting the alpha coefficient (refer to Figure 9). However, for a more detailed analysis, we also experimented to observe how alpha coefficients vary according to different model sizes, training chat data, and architectures. For instance, when we used LLAMA-2-13B Chat instead of LLAMA-2-7B Chat to explore changes according to model size, the alpha coefficient remained constant at 0.4. Additionally, we replaced LLAMA-2-7B Chat with the base model LLAMA-2-7B, which was not tuned with a chat dataset, and found that the alpha coefficient still maintained at 0.4.
>
> Lastly, to examine changes according to architecture, we replaced LLAMA-2 Chat 7B with Mistral-7B-Instruct-v0.3, resulting in an alpha coefficient shift from 0.4 to 0.5. Furthermore, we applied a model stock merging with different merging ratios at each layer [1], which, compared to linear merging, resulted in lower performance, thereby justifying our choice of linear merging in this work.
>
> |Ablation|Alpha coefficient (applied to the SL)|
> |-----------|------------------------------------------------|
> |LLaMA-2 Chat 7B -> LLaMA-2 Chat 13B|0.4|
> |LLaMA-2 Chat 7B -> LLaMA-2 7B|0.4|
> |LLaMA-2 Chat 7B -> Mistral 7B Instruct v0.3|0.5|
>
> |Merging method|Text-only ASR ↓|Multimodal ASR  ↓|Multimodal helpfulness ↑|
> |----------------------|-------------------|---------------------|-------------------------------|
> |Linear Merging|0.31|0.72|61.6|
> |Model Stock Merging|1.13|3.82|60.9|
>
> ---
>
> **Trade-off between safety and effectiveness (W5)**
>
> In this work, we acknowledge that model weight merging has not completely resolved the trade-off relationship, and as you mentioned, it primarily demonstrates balanced performance. However, another study [2] has shown that weight merging can achieve higher performance on two distinct tasks than models trained on each task separately. This suggests that with further advancements in merging techniques and the development of more robust safety datasets, it might be possible to resolve the trade-off relationship in the future.

---

> > ### Author Response · Authors · 2024-11-19
> > **Rebuttal by Authors (2)**
> >
> > **Theoretical or technical explanations for balancing (W6)**
> >
> > Thank you for your insightful comment. Multi-task learning seeks to share knowledge across different tasks, enhancing data efficiency and overall generalization. However, training simultaneously on multiple tasks can lead to optimization issues, resulting in performance that might not match that of independently trained tasks. [3] outlines three primary causes for these optimization challenges:
> >
> > 1. Conflicting Gradients, where task gradients oppose each other, impeding progress; 2. Gradient Domination, where the gradient magnitude of one task overwhelmingly influences the optimization path; 3. High Curvature, which presents difficulties for optimizers in navigating steep regions of the loss landscape.
> >
> > They propose PCGrad (Project Conflicting Gradients) to mitigate these issues, which has shown to improve convergence, especially under conditions of significant gradient conflicts and high curvature. Inspired by these findings, we hypothesized that the suboptimal performance in multi-task learning scenarios, such as those involving safety tuning and VL adaptation, is due to these mentioned challenges. Our rationale stems from the fact that safety tuning typically trains models to refuse responses to harmful questions, whereas VL adaptation focuses on accurately understanding and responding to user queries.
> >
> > ---
> >
> > **Filtering quality (Q1)**
> >
> > Inspired by [4], we have implemented a filtering process for LLaVA-pretrain and LLaVA-Instruct, utilizing LLaMA-Guard-3 8B [5] to filter harmful text and an NSFW image detection model [6] to filter unsafe images. LLaMA-Guard-3 8B is a classifier that deems text as unsafe if it includes any of 14 specified harmful elements. Out of a total of 664,943 data entries, 609 texts, representing 0.09%, contained harmful elements. Similarly, the NSFW image classifier assesses images for inappropriate content, identifying 7 images, or 0.001% of the total data, as containing harmful elements. Consequently, 612 examples were filtered out, leaving 664,331 entries for use as our final training dataset.
> >
> > ---
> >
> > **Evaluation metric (Q2)**
> >
> > Thank you for pointing out the error. The metric in question is actually higher is better, not lower is better. We will correct this in the revised version.
> >
> > ---
> >
> > **References**
> >
> > [1] Model Stock: All we need is just a few fine-tuned models (Jang et al., ECCV 2024)
> >
> > [2] Prometheus 2: An Open Source Language Model Specialized in Evaluating Other Language Models (Kim et al., EMNLP 2024)
> >
> > [3] Gradient Surgery for Multi-Task Learning (Yu et al., NeurIPS 2020)
> >
> > [4] WildVision: Evaluating Vision-Language Models in the Wild with Human Preferences (Lu et al., 2024)
> >
> > [5] https://www.llama.com/docs/model-cards-and-prompt-formats/llama-guard-3/
> >
> > [6] https://huggingface.co/Falconsai/nsfw_image_detection

---

> > > ### Author Response · Authors · 2024-11-25
> > > **Kindly reminder for discussion**
> > >
> > > Dear Reviewer 2icJ, We appreciate for your time and effort on reviewing the paper. As we near the end of the discussion period, we kindly remind you of the upcoming deadline. We are eager to discuss any aspects of the paper that may require further clarification. Thank you once again for your valuable feedback.

---

> > > > ### Comment · Reviewer_2icJ · 2024-11-25
> > > >
> > > > Thanks for your whole efforts to response my questions.
> > > > W1, W2, Thank you. While the additional experiments address some variation in base models and configurations, the coverage remains insufficient to establish generalizability across the broader ecosystem of vision-language models. Key models such as Qwen-VL-2B, ShareGPT4V-13B, and Qwen2-VL-72B, which represent significant diversity in training paradigms and parameter scales, are still not included. Including these models would significantly strengthen the claim of universal trends in safety degradation. It’s another issue if you have difficulties on availability of large-scale computing power.
> > > >
> > > > The ablation studies on vision encoders and connectors are appreciated, but the scope is limited. Using only variations of OpenAI-LargeCLIP and a single linear layer as alternatives fails to capture the diversity in encoder and connector architectures present in current state-of-the-art models. Experimenting with a broader range of encoders and connectors (e.g., attention-based mechanisms) would provide more robust evidence of generalizability.
> > > >
> > > > W3, W4, Thank you. While the authors claim the uniqueness of applying weight merging to multimodal safety, they do not provide sufficient evidence to demonstrate that this represents a significant methodological innovation over existing approaches such as TIES-Merging. Without further theoretical or experimental justification for the specific challenges of multimodal safety addressed by weight merging, this contribution risks being perceived as an incremental application rather than a novel methodological advancement. The additional experiments are appreciated; however, the evaluation remains limited to specific architectures (LLaMA and Mistral) and scales (up to 13B). To strengthen the claim of broad applicability, further validation on a wider range of architectures (e.g., Qwen-VL, Bloom, OPT) and training paradigms is necessary. Additionally, the experiments should test varying datasets and task types to ensure robustness across diverse multimodal challenges. While the observed stability of the alpha coefficient is interesting, it remains unclear why such consistency arises across different models and whether this trend holds for larger or smaller models (e.g., 34B, 72B, or 2B). Without further theoretical or empirical analysis, the claim of alpha stability risks being premature and context-dependent.
> > > >
> > > > W6, Thanks you. While the authors reference optimization challenges such as conflicting gradients and gradient domination, the discussion lacks specificity regarding how these issues manifest uniquely in the context of safety tuning and multimodal tasks. A more rigorous mathematical framework or empirical analysis quantifying these conflicts would significantly strengthen the theoretical grounding of the paper's conclusions.
> > > >
> > > > Q1, Thank you. The filtering process removed only a negligible proportion of the data (0.09% of text and 0.001% of images), raising questions about whether such a small adjustment has any meaningful impact on improving the safety of the dataset. Further analysis is needed to validate the significance of these filtered examples on model performance and safety.

---

### Official Review · Reviewer_WnEH · 2024-11-02

**Soundness:** 3
**Presentation:** 3
**Contribution:** 3
**Rating:** 6
**Confidence:** 4

**Summary:**

This paper examines how the vision-language adaptation process influences model safety. By conducting safety fine-tuning on Llama and Tulu models and evaluating their safety across various benchmarks, the authors draw several conclusions: 1) Safety degradation occurs during adaptation, even when the training data is safe. 2) SFT and RLHF lead to safety degradation and reduced helpfulness due to over-rejection. 3)VL adaptation may affect certain safety-related layers, lowering overall safety. 4) The objectives of adaptation and safety tuning are not aligned, resulting in suboptimal outcomes for both.

Additionally, this paper proposes using weight merging to reduce safety degradation while maintaining the model's helpfulness.

**Strengths:**

The paper addresses an important topic, providing insightful analysis that holds good value for the community. The authors conduct comprehensive experiments, adding depth and rigor to their findings

**Weaknesses:**

The theoretical analysis could benefit from greater depth, particularly regarding the roles of individual modules within a block, such as comparing the importance of the FNN versus attention components, and examining whether this knowledge could support model merging. Additional analysis on why model merging results in a better trade-off would also be valuable. For instance, investigating whether layer-wise divergence decreases after model merging, compared to fine-tuning, could offer insights into the underlying mechanisms.

It is also noteworthy that model merging yields the lowest multimodal helpfulness in Table 4, raising the question of whether this approach achieves an optimal trade-off. If higher multimodal helpfulness is a priority for the community, could an improved model merging strategy provide a better balance?

Furthermore, a more detailed comparison of different tuning methods would enrich the analysis. For example, which fine-tuning method, such as SFT or RLHF, strikes a better balance between safety and performance?

Lastly, all experiments in the paper are based on Llama models. While replicating the experiments across different LLMs may involve considerable cost, understanding whether similar conclusions apply to other foundational models would greatly benefit the community, as variations across LLMs could influence the generalizability of the findings.

**Questions:**

- What are the specific attacking methods used? My current impression is that there is no optimization-based attacking methods used in this study. If adversarial training were considered, would the conclusions remain similar?
- It would be beneficial to move the main experiment table on model merging to the main paper, rather than placing it in the appendix.

---

> ### Author Response · Authors · 2024-11-19
> **Rebuttal by Authors (1)**
>
> Dear reviewer WnEH, we deeply appreciate your constructive feedback. Here we address the weaknesses (W) and questions (Q) that you raised on our work.
>
> **Further analysis on model weight merging (W1)**
>
> To visually analyze the balanced performance of linear merging, we conducted the experiment you mentioned, following the same setup as described in Section 5.2. Please refer to Figure 12 in the revised version of the attached paper for the experimental results plot. The results confirmed that the layer-wise divergence between the linear merged model and the VL model was located between the chatty and SL models, which were the targets of the merging. This observation demonstrates that linear merging effectively combines the behaviors of the two models. It supports that the performance of the linear merging model is indeed balanced, corroborating its effectiveness in blending different model characteristics.
>
> ---
>
> **Some discussions on comparison between SFT and RLHF (W2)**
>
> First, our analysis of the results in Table 1 and Figure 6 confirms that applying RLHF does not ensure safety to the same extent in multimodal models as it does in LLMs. Furthermore, a qualitative analysis in Appendix F revealed that while models trained with SFT respond more appropriately to harmful questions, those trained with RLHF perform better on questions that are not harmful. This suggests that while the debate over which method—SFT or RLHF—is more effective in achieving balanced performance is somewhat resolved in LLMs, it remains unsettled in multimodal models.
>
> Generally, in the LLM domain (non-multimodal), it is well-documented that using RLHF for safety tuning successfully creates AI that is both helpful and safe without compromising other functionalities. However, the situation on the multimodal side is less clear. This ambiguity is largely due to the still insufficient RLHF training datasets and reward models dedicated to safety. While most multimodal RLHF datasets are employed to reduce hallucinations and enhance multimodal helpfulness [1, 2], the datasets specifically concerning safety [3] are limited in quantity. Due to these factors, achieving a balance between safety and helpfulness in multimodal models remains a challenging task, which we plan to address in future work.
>
> ---
>
> **Model architecture ablations (W3)**
>
> In Table 4, we report the results of experiments conducted by replacing the base LLM with Vicuna, and in Table 5, we measure safety and multimodal helpfulness for other models available as open-source, including LLaVA Next 8B, Phi-3-Vision-Instruct, and MiniCPM-V-2.6, besides LLaVA v1.5. These results demonstrate that despite changing the base LLM and using different VLMs, the safety issues remain unresolved, illustrating that the safety degradation phenomenon we are addressing in this work is widespread. Additionally, we have included results from VL adaptations using LLaMA-2 7B, LLaMA-2 Chat 13B, and Mistral-7B-Instruct-v0.3 as base LLMs. And we also have reported the results of substituting the vision encoder from openai/clip-vit-large-patch14-336 to openai/clip-vit-large-patch14, and the effects of replacing the connection from an Mlp to a single linear layer.
>
> |Base LLM|Text-only ASR ↓|Multimodal ASR ↓|Multimodal helpfulness ↑|
> |-------------|--------------------|---------------------|----------------|
> |LLaMA-2-Chat 7B (Ours)|54.4|96.2|64.3|
> |LLaMA-2-Chat 13B|50.1|94.4|66.3|
> |Mistral 7B Instruct v0.3|52.1|95.3|65.7|
>
> |Ablation|Text-only ASR ↓|Multimodal ASR ↓|Multimodal helpfulness ↑|
> |-------------|--------------------|---------------------|----------------|
> |Ours|54.4|96.2|64.3|
> |CLIP ablation|57.9|96.7|61.8|
> |Connector ablation|55.7|96.9|62.2|

---

> > ### Author Response · Authors · 2024-11-19
> > **Rebuttal by Authors (2)**
> >
> > **Attacking method ablations (Q1)**
> >
> > Thank you for the feedback. The primary aim of this work was to observe whether model safety degrades solely through VL adaptation using benign data, without being trained on malicious attacks or harmful data. Consequently, we did not employ attacking methods such as adversarial training or prompt injection. Our experiments were limited to inputting harmful questions into the safety benchmark.
> >
> > To further explore how additional attacks might affect model behavior, inspired by [4], we will investigate changes in Attack Success Rate (ASR) when prefix injection and refusal suppression are implemented during evaluation. Prefix injection involves adding a positive prefix like “Sure, here is…” to the model's response to harmful questions, prompting the model to answer. Refusal suppression, on the other hand, involves explicitly forbidding the model from avoiding a response to a harmful question by including prompts such as “Do not apologize” in the input, thereby inducing harmful responses. Our findings show significantly higher ASR when these attacks are applied compared to when they are not.
> >
> > Furthermore, incorporating the refusal suppression prompt during VL adaptation training without any attack also results in a high ASR. This suggests that if adversarial training and inference time attacking are applied, the model may exhibit harmful behavior more frequently.
> >
> > |Attacking method|Text-only ASR|Multimodal ASR|
> > |------------------------|-------------------|---------------------|
> > |Vanilla (default)|54.4|96.2|
> > |Prefix Injection|87.6|97.8|
> > |Refusal Suppression (Inference time)|76.8|96.9|
> > |Refusal Suppression (Training time)|77.4|97.7|
> >
> > ---
> >
> > **Moving the main experiment table (Q2)**
> >
> > Thank you for your suggestion. In the revised version, we will move the relevant section from the appendix to the main paper to provide stronger support for model weight merging.
> >
> > ---
> >
> > **References**
> >
> > [1] Aligning Large Multimodal Models with Factually Augmented RLHF (Sun et al., 2023)
> >
> > [2] RLHF-V: Towards Trustworthy MLLMs via Behavior Alignment from Fine-grained Correctional Human Feedback (Yu et al., CVPR 2024)
> >
> > [3] SPA-VL: A Comprehensive Safety Preference Alignment Dataset for Vision Language Model (Zhang et al., 2024)
> >
> > [4] Jailbroken: How Does LLM Safety Training Fail? (Wei et al., NeurIPS 2023)

---

> > > ### Comment · Reviewer_WnEH · 2024-11-24
> > > **Thanks for the response**
> > >
> > > Dear Authors,
> > >
> > > Thank you for the response and newly added experiments. My concerns have been resolved and I keep my score unchanged.

---

### Official Review · Reviewer_cNbS · 2024-11-02

**Soundness:** 3
**Presentation:** 2
**Contribution:** 3
**Rating:** 6
**Confidence:** 3

**Summary:**

This paper addresses the degradation of safe behaviors in large language models (LLMs) when they are fine-tuned for vision-language (VL) tasks using the widely adopted LLaVA approach. It compares different methods to enhance safety in these models, including variations of supervised fine-tuning (SFT) and direct preference optimization (DPO). The authors propose merging model weights to improve both safety and helpfulness, aiming to find a balance between these crucial aspects in VL models.

**Strengths:**

- *Important Problem Addressed*: The paper tackles the critical issue of safety degradation in LLMs when adapted for VL tasks, which is essential for the responsible deployment of AI systems.
- *Comparative Analysis*: It provides a thorough comparison of various approaches to increase safety in VL models, offering valuable insights into their effectiveness.
- *Novel Methodology*: The introduction of model weight merging as a technique to balance safety and helpfulness is innovative and contributes to the field.
- *Empirical Evaluation*: The paper includes experiments using established benchmarks to evaluate safety and helpfulness, providing quantitative data to support the proposed methods.

**Weaknesses:**

(by order of importance)
- *Lack of Clear Definitions*: The concepts of safety and helpfulness are central to the paper but are not explicitly defined or discussed in depth. The reliance on benchmarks offers a precise but narrow implicit definition, limiting the broader understanding of these terms.
- *Insufficient Exploration of Main Contribution*: The weight merging technique, being a core novelty, lacks sufficient experiments and detailed analysis. For instance, exploring whether the same alpha value is optimal across different model sizes and architectures would strengthen the contribution.
- *Omission of Relevant Work*: The paper does not discuss related studies such as Prismatic-VLM, which explored co-training for VL and text safety SFT and showed benefits in image safety through cross-modal transfer.
- *Incomplete Experimental Details*: Key experimental details are missing, such as the extent and criteria of filtering unsafe examples from the llava-pretrain and llava-instruct datasets, and assessments of the filtering technique used (line 164), or the data mixing ratio used in section 4.2.
- *Appendix Dependence*: Essential components are relegated to the appendix, leading to an incomplete understanding of the methods and results in the main text. (Figure 9 could be included in the main part of the paper for example). If the author could not fit the whole content in 10 pages, I would suggest pushing section 5.1 to the appendix as it does not convincingly support the claims of the paper.
- *Presentation Issues*: Figures 2 and 3 have dual y-axes with similar but unaligned scales, causing confusion. Aligning these scales would improve clarity.
- *Overstated Claim*: The introduction claims to "enhance both safety and multimodal capabilities," but the method primarily finds a compromise between the two rather than improving both simultaneously.

**Questions:**

related to weaknesses above
- Could you provide clear definitions and discussions of safety and helpfulness in the context of your work? How do the benchmarks used capture these concepts, and what are their limitations?
- Have you considered experimenting with different alpha values per layer in your weight merging method, especially given the identification of "safety layers"? Is the same alpha value optimal across different model sizes, training pipelines and architectures (e.g., Mistral, larger models, base models, chat models)?
- Co-training with text-only safety data is left mainly unexplored while it would be an interesting point of comparison. In section 4, does the merging alpha coefficient relate in any way to the data mix that one could use to train a safe and helpful model?
- Could you elaborate on the criteria used for filtering and quantify how much content was removed? Was the filtered content genuinely unsafe? What data mixing ratio was used in section 4.2?
- In lines 385-390, the sentence structure is unclear. The appendix suggests that you observe safety degradation regardless of which group of layers was frozen during your experiments, is that what is meant?

**Details Of Ethics Concerns:**

The paper gives examples of unsafe model answers which help understand the subject but are clearly indicated as potentially "distressing or offensive".

---

> ### Author Response · Authors · 2024-11-19
> **Rebuttal by Authors (1)**
>
> Dear reviewer cNbS, we deeply appreciate your constructive feedback. Here we address the weaknesses (W), questions (Q) and ethics concerns (E) that you raised on our work.
>
> **Ethics concerns about unsafe contents in paper (E1)**
>
> In Appendix F, where harmful examples are qualitatively analyzed, we included a warning message at Line 1114 to inform readers that this section may contain distressing or offensive material. However, recognizing that this might not be sufficient, we have implemented blurry bounding on offensive and distressing content to address ethical concerns. A revised version of the paper reflecting these changes has been uploaded.
>
> ---
>
> **Clear definitions and discussions of safety and helpfulness (W1, Q1)**
>
> In this work, we define safety in two distinct contexts. The first is text-only/multimodal safety, which refers to the model’s ability to avoid generating harmful responses when presented with harmful inputs. These inputs include both text-only scenarios (e.g., SorryBench [1], WildJailbreak [2]) and multimodal scenarios (e.g., FigStep [3], MM-SafetyBench [4], SIUO [5]). The second is exaggerated safety, evaluated using XSTest [6], which measures the model’s ability to avoid overly safe responses when presented with benign inputs.
>
> Helpfulness, on the other hand, is specifically defined in terms of multimodal helpfulness. This refers to the ability of a large vision-language model (LVLM) to understand vision-language inputs and provide accurate, helpful answers to user queries (e.g., MMBench [7], MME [8], SEEDBench [9]). These evaluations are critical because an LVLM must not only generate harmless responses but also provide responses that align with user intent to remain effective and ethically compliant.
>
> The reason we conducted evaluations across these two dimensions is that it is essential for LVLMs to strike a balance between ethical responsibility and utility, generating harmless yet helpful responses. The benchmarks we used are well-established in previous works and are effective for assessing the definitions of safety and helpfulness outlined in our study. However, we acknowledge certain limitations in these benchmarks. Specifically, multimodal benchmarks lack evaluation samples for NSFW images due to the constraints of data collection. This poses challenges when models encounter such inputs, which highlights a limitation in these benchmarks.
>
> ---
>
> **Weight merging ablations (W2, Q2)**
>
> We conducted ablation studies that involved applying different merging methods beyond linear merging and varying the combinations of models targeted for merging (see Table 3), as well as adjusting the alpha coefficient (refer to Figure 9). However, for a more detailed analysis, we also experimented to observe how alpha coefficients vary according to different model sizes, training chat data, and architectures. For instance, when we used LLAMA-2-13B Chat instead of LLAMA-2-7B Chat to explore changes according to model size, the alpha coefficient remained constant at 0.4.
>
> Additionally, we replaced LLAMA-2-7B Chat with the base model LLAMA-2-7B, which was not tuned with a chat dataset, and found that the alpha coefficient still maintained at 0.4. Lastly, to examine changes according to architecture, we replaced LLAMA-2 Chat 7B with Mistral-7B-Instruct-v0.3, resulting in an alpha coefficient shift from 0.4 to 0.5. Furthermore, we applied a model stock merging with different merging ratios at each layer [10], which, compared to linear merging, resulted in lower performance, thereby justifying our choice of linear merging in this work.
> |Ablation|Alpha coefficient (applied to the SL)|
> |-----------|------------------------------------------------|
> |LLaMA-2 Chat 7B -> LLaMA-2 Chat 13B|0.4|
> |LLaMA-2 Chat 7B -> LLaMA-2 7B|0.4|
> |LLaMA-2 Chat 7B -> Mistral 7B Instruct v0.3|0.5|
>
> |Merging method|Text-only ASR ↓|Multimodal ASR  ↓|Multimodal helpfulness ↑|
> |----------------------|-------------------|---------------------|-------------------------------|
> |Linear Merging|0.31|0.72|61.6|
> |Model Stock Merging|1.13|3.82|60.9|

---

> > ### Author Response · Authors · 2024-11-19
> > **Rebuttal by Authors (2)**
> >
> > **Co-training with text-only safety data (W3, Q3)**
> >
> > Thank you for your valuable suggestion. Initially, we conducted co-training using text-only safety data (referred to as MTL in our study), which showed positive effects on the text-only safety benchmark but did not perform as well on the multimodal safety benchmark. However, when we included multimodal safety data in our co-training, the results were not as significant as those from SL and model merging, but it still showed improvements on both text-only and multimodal safety benchmarks. This is why we opted to use multimodal safety data in this work.
> >
> > Additionally, as you mentioned, we explored the relationship between the data mix ratio and the alpha coefficient used during model merging. We adjusted the ratio of safety data to vl adaptation data used in co-training to 4:6 (default setting is 3:7) for this purpose. Unfortunately, due to resource limitations, we were unable to perform ablation studies across all scenarios, ranging from a 1:9 to a 9:1 ratio. Still, co-training did not show improved performance, leading us to hypothesize that model weight merging might not be directly related to data mixing. Model weight merging occurs after each model has independently completed training on its task, likely minimizing interference from other tasks. In contrast, co-training through data mixing potentially introduces interference during parameter updates due to the presence of different tasks.
> >
> > |safety data|Text-only ASR ↓|Multimodal ASR ↓|
> > |---------------|-------------------|--------------------------|
> > |MTL w/ Multimodal safety data (Ours)|44.8|3.1|
> > |MTL w/ Text-only safety data|45.2|33.2|
> >
> > |Data mixing / weight merging ratio (safety:vl)|Text-only ASR ↓|Multimodal ASR ↓|Multimodal helpfulness ↑|
> > |-------------------------------------|-------------------|--------------------------|---------------------------------|
> > |Data mixing 4:6|43.1|2.5|60.3|
> > |Linear Merging 4:6|0.31|0.72|61.6|
> >
> > ---
> >
> > **Data filtering (W4, Q4)**
> >
> > Inspired by [11], we have implemented a filtering process for LLaVA-pretrain and LLaVA-Instruct, utilizing LLaMA-Guard-3 8B [12] to filter harmful text and an NSFW image detection model [13] to filter unsafe images. LLaMA-Guard-3 8B is a classifier that deems text as unsafe if it includes any of 14 specified harmful elements. Out of a total of 664,943 data entries, 609 texts, representing 0.09%, contained harmful elements. Similarly, the NSFW image classifier assesses images for inappropriate content, identifying 7 images, or 0.001% of the total data, as containing harmful elements. Consequently, 612 examples were filtered out, leaving 664,331 entries for use as our final training dataset.
> >
> > ---
> >
> > **Clarification on results of layer freezing (Q5)**
> >
> > In Appendix D.3 and Table 6, we aimed to demonstrate two key points. First, we wanted to show that freezing the specific safety layer, as opposed to freezing other layers or not freezing any at all and training the entire model (LLaMA-2-Chat-VL), offers benefits on safety benchmarks. This supports our claim in Section 5.1 that safety degradation is associated with the safety layer. Secondly, despite these benefits, our results still show lower safety performance compared to baselines that explicitly underwent safety tuning (MTL, SL, SafeRLHF, SPA-VL). This indicates that freezing merely helps to preserve the original model’s safety knowledge without undergoing VL adaptation, and emphasizes the need for proactive safety tuning to achieve optimal safety performance.
> >
> > ---
> >
> > **Paper structure and over-claimimg issues: (W5, W6, W7)**
> >
> > Thank you for your valuable feedback. As you suggested, we will move the content from Section 5.1 to an appendix and relocate the ablation studies, currently in the appendix, to the main paper. This adjustment will better support our method and the analyses we wish to emphasize. Additionally, we will revise Figures 2 and 3, along with the overall layout of the figures, to minimize confusion. Finally, in line with your recommendation, we will modify the phrase "enhance both safety and multimodal capabilities" in the introduction to convey a more balanced enhancement of safety and multimodal capabilities, thus avoiding any overclaims.

---

> ### Author Response · Authors · 2024-11-19
> **Rebuttal by Authors (3)**
>
> **References**
>
> ---
>
> [1] SORRY-Bench: Systematically Evaluating Large Language Model Safety Refusal Behaviors (Xie et al., 2024)
>
> [2] WildTeaming at Scale: From In-the-Wild Jailbreaks to (Adversarially) Safer Language Models (Jiang et al., 2024)
>
> [3] FigStep: Jailbreaking Large Vision-language Models via Typographic Visual Prompts (Gong et al., 2023)
>
> [4] MM-SafetyBench: A Benchmark for Safety Evaluation of Multimodal Large Language Models (Liu et al., 2023)
>
> [5] Cross-Modality Safety Alignment (Wang et al., 2024)
>
> [6] XSTest: A Test Suite for Identifying Exaggerated Safety Behaviours in Large Language Models (Rottger et al., NAACL 2024)
>
> [7] MMBench: Is Your Multi-modal Model an All-around Player? (Liu et al., ECCV 2024)
>
> [8] MME: A Comprehensive Evaluation Benchmark for Multimodal Large Language Models (Fu et al., 2024)
>
> [9] SEED-Bench: Benchmarking Multimodal LLMs with Generative Comprehension (Li et al., 2023)
>
> [10] Model Stock: All we need is just a few fine-tuned models (Jang et al., ECCV 2024)
>
> [11] WildVision: Evaluating Vision-Language Models in the Wild with Human Preferences (Lu et al., 2024)
>
> [12] https://www.llama.com/docs/model-cards-and-prompt-formats/llama-guard-3/
>
> [13] https://huggingface.co/Falconsai/nsfw_image_detection

---

> > ### Comment · Reviewer_cNbS · 2024-11-23
> >
> > About the ethic statement, thank you for taking these measures. I mainly flagged this in the hope to have someone more competent than myself in the matter from the ICLR committee have a look, not really for you to sensor your paper. I should have made this clearer but I still appreciate the steps taken.
> >
> > W1
> > Thank you, I understand what the benchmarks measure and the trade-off between usefulness and harmfulness. However, when considering safety, I think it is important to go further than the strict and narrow benchmark measurement definition and discuss harmfulness that is not measured by the benchmarks even if it would become more philosophical and less measurable. I believe it was useful to include some of the harmful content to give a qualitative sense of what it means. Thus you covered the qualitative, the measures but not the desired definition (for example one may consider a strong political bias as harmful and this is not measured here).
> > I know this is out of scope for this rebuttal and maybe for your work but partnering with a philosopher on the harmfulness question (or simply citing some of this literature) and having a profound analysis of the matter and the distance from its concept to your measured results and how it relates to the improvement you propose would make this paper a strong accept in my opinion.
> >
> > W2 The fact that the same (or close) alpha value is empirically best in so many settings is very interesting. Thank you for these details.
> >
> > W3 These results are very intriguing and interesting for the community.
> >
> > W4 Thank you, mentioning that it is around 0.1% of the data is important. Qualitatively, did you find these filtered element harmful?
> >
> > You have answered most of my questions and the rest might be out of scope, thank you.

---

> > > ### Author Response · Authors · 2024-11-25
> > > **Official Comment by Author**
> > >
> > > Thank you for the thoughtful comment! Regarding your question about W4, we observed that the 0.1% of examples in question often contained text that was harmful or exhibited bias, even from a qualitative perspective. Notably, most of these examples were related to bias, and if not filtered, there was a potential risk that the model could learn from such content.

---

### Official Review · Reviewer_EKaT · 2024-11-04

**Soundness:** 3
**Presentation:** 3
**Contribution:** 3
**Rating:** 8
**Confidence:** 4

**Summary:**

This paper explores the impact of Vision-Language (VL) adaptation on the safety capabilities of Large Language Models (LLMs) when transformed into Large Vision-Language Models (LVLMs). The study reveals that VL adaptation significantly compromises safety measures, even with carefully filtered training data, due to the disruption of key safety-related layers and conflicting objectives between safety tuning and VL adaptation. The authors propose a model weight merging approach to address these challenges, achieving a better balance between safety and functionality without requiring extensive retraining.

**Strengths:**

- Clear presentation. Figure 1 explains different safety type and role of this paper in it very clearly
- Designed comprehensive evaluation metrics
-

**Weaknesses:**

- Figure 6 does not show safety score baseline of non multimodal LLaMA-2-Chat-7B which would be very helpful as comparison.
- Lacks more in-depth of analysis of what exactly changes during finetune that alter model behaviors. The paper focuses more on end-2-end intervention. But some more mechanistic understanding of training dynamics would be interesting.
- Lacks ablation on VL and safety tuning datasets.

**Questions:**

- In figure 2, what's the motivation to show number of steps > 2000? There seems not many changes in metrics across steps.
- Section 5.1 shows that weight's similarity to safety layers decrease along with training. What do these layers change to? Are they repurposed to processing other features or gain some non-safe behaviors?
- Since Section 5.1 identifies safety layers, can we freeze some parameters while doing VL adaptations? Another baseline would be mixing safety data while doing VL adaptation.
- Would be very helpful to further contextualize the paper in recent literatures that examine weight merging as a way to mitigate non-multimodal's safety mechanisms after finetune. What makes the problem of multimodal model special?
- What should audience conclude from Figure 4?

---

> ### Author Response · Authors · 2024-11-19
> **Rebuttal by Authors (1)**
>
> Dear reviewer EKaT, we deeply appreciate your constructive feedback. Here we address the weaknesses (W) and questions (Q) that you raised on our work.
>
> **Add LLaMA-2-Chat-7B Performance to Figure 6 (W1)**
>
> Thank you for the suggestion. We will include the safety performance of LLaMA-2-Chat 7B in Figure 6 and incorporate this update in the revised version.
>
> ---
>
> **More analysis about training dynamics (W2)**
>
> Thank you for the great suggestion. To better understand the model's behavior during training, we conducted additional validation every 2000 steps during the VL adaptation process using a safety benchmark. This benchmark includes a harmful question along with a Chosen response and a Rejected response. During training, we measured the Chosen Loss and Rejected Loss at each validation point. Refer to Figure 11 in the revised version of the attached paper for the experimental results plot.
>
> By subtracting the Chosen Loss from the Rejected Loss, we observed that the difference between the two losses decreased as training progressed. Since loss represents the negative average log likelihood of the predicted tokens for a given input, a lower loss indicates that the model is more likely to generate responses closer to the expected answer. For a safe model, the Chosen Loss should be smaller than the Rejected Loss because the model should preferentially generate the safer Chosen responses.
> However, as shown in the plot, while the Rejected Loss remains higher than the Chosen Loss throughout training, the difference between the two narrows as VL adaptation progresses. By the end of VL adaptation, the difference becomes minimal. This observation suggests that VL adaptation causes the model to become more inclined to generate harmful responses, undermining its safety.
>
> ---
>
> **Training data ablation (W3)**
>
> We conducted ablation studies to examine whether our results remain consistent across various training datasets and those used for safety tuning. Initially, we replaced the VL adaptation data from LLaVA v1.5 with vision-flan data [1]. This change led to a slight improvement in multimodal performance; however, the text-only ASR and multimodal ASR remained high. Secondly, we substituted the multimodal safety tuning data with text-only safety tuning data [2] and observed the results. While this modification was effective in reducing text-only ASR, it did not yield good results for multimodal ASR. These findings demonstrate that the phenomenon of safety degradation in VLMs occurs regardless of the learning data used, supporting the assertion that VL adaptation negatively impacts safety without proactive safety tuning.
>
> |VL data|Text-only ASR ↓|Multimodal ASR ↓|Multimodal helpfulness ↑|
> |---------------|-------------------|--------------------------|------------------------------|
> |LLaVA v1.5 data (Ours)|54.4|96.2|64.3|
> |Vision Flan|60.1|97.1|65.7|
>
> |safety data|Text-only ASR ↓|Multimodal ASR ↓|Multimodal helpfulness ↑|
> |---------------|-------------------|--------------------------|------------------------------|
> |MTL w/ Multimodal safety data (Ours)|44.8|3.1|64.1|
> |MTL w/ Text-only safety data|45.2|33.2|62.8|
>
> ---
>
> **Motivation to show number of steps > 2000 (Q1)**
>
> It is true that there are no significant changes after 2000 steps; however, 2000 steps only represent the early stages of the entire fine-tuning process. Even if the changes are minimal, we decided to present the results for the full fine-tuning steps to address any potential concerns or doubts readers might have if the complete results were not shown.

---

> ### Author Response · Authors · 2024-11-19
> **Rebuttal by Authors (2)**
>
> **Clarification on safety layers (Q2, Q3, Q5)**
>
> In the original introduction of safety layers by [1], they are described as “a small set of contiguous layers in the middle of the model that are crucial for distinguishing malicious queries from normal ones,” demonstrating that freezing and fine-tuning only these layers can improve safety performance. Inspired by this, we observed that when harmful inputs are given to models with and without VL adaptation, the cosine similarities between the hidden states output by the safety layer decrease. This indirectly suggests that one reason for the safety degradation caused by VL adaptation is the change in the parameters of the safety layer.
>
> We present these results in Figure 4. After VL adaptation, the cosine similarities between the output hidden states of the safety layer when given harmful questions and those of the safety layer before VL adaptation are as low as 0.5. Through this, we aimed to provide readers with the insight that VL adaptation alters the weights of the safety layer, leading to safety degradation. While the cosine similarities decrease even further in later layers, we think these layers play less critical roles in safety, meaning their impact on safety performance is likely minimal. This assumption is further supported by the layer freezing experiments discussed later, which validate our claim.
>
> Furthermore, as shown in Figure 6, we applied the safely partial parameter fine-tuning (SPPFT) method [3], which involves freezing only the safety layers during VL adaptation. The results indicate improved safety performance compared to when the safety layers were not frozen. Additionally, Table 6 in Appendix D.3 of our work provides additional evidence by demonstrating that freezing parameters in layers other than the safety layer does not lead to improvements in safety performance. These findings support our argument regarding the critical role of the safety layer in maintaining safety performance during adaptation.
>
> ---
>
> **Model weight merging in multimodal models (Q4)**
>
> Thank you for your feedback. While there are previous works that apply weight merging to non-multimodal models to achieve well-rounded performance across various tasks, including safety, most of them focus on improving performance in specific knowledge or reasoning tasks rather than prioritizing safety. To the best of our knowledge, there has been no prior work addressing multimodal safety in this context.
>
> What makes model weight merging particularly significant for multimodal models is the unique nature of these models. Unlike non-multimodal models, multimodal models process multiple modalities simultaneously (in this work, both images and text), which results in significantly higher training costs. We believe that applying weight merging in this context has the potential to be a much more efficient approach for multimodal models, given their higher computational demands.
>
> ---
>
> **References**
>
> [1] Vision-Flan: Scaling Human-Labeled Tasks in Visual Instruction Tuning (Xu et al., 2024)
>
> [2] WildGuard: Open One-Stop Moderation Tools for Safety Risks, Jailbreaks, and Refusals of LLMs (Han et al., 2024)
>
> [3] Safety Layers in Aligned Large Language Models: The Key to LLM Security (Li et al., 2024)

---

> > ### Comment · Reviewer_EKaT · 2024-11-19
> >
> > Thanks for addressing my concerns. The supporting evidence and arguments in rebuttal resolved my questions. I maintain my decision of acceptance.

---

### Official Review · Reviewer_wpWa · 2024-11-04

**Soundness:** 3
**Presentation:** 3
**Contribution:** 3
**Rating:** 6
**Confidence:** 3

**Summary:**

The paper investigates how vision-language adaptation affects the safety of vision-language models. It demonstrates that VL adaptation degrades the inherent safety capabilities of large language models, even with safe training data. The study shows that existing safety fine-tuning methods like supervised fine-tuning and RLHF mitigate safety issues but will reduce the model's helpfulness. The authors propose a weight merging approach as a solution to balance safety and helpfulness in LVLMs.

**Strengths:**

1. Provides a comprehensive analysis of safety degradation during VL adaptation.
2. Utilizes various safety and multimodal benchmarks.
3. Introduces model weight merging as a cost-effective way to balance safety and performance.

**Weaknesses:**

1. While the proposed solutions mitigate safety degradation, they may still lead to trade-offs in multimodal performance that require more precise quantification. Notably, the linear merged model exhibits significantly lower accuracy. Would more selective merging strategies, such as conditioning on whether a layer is a safety-related layer, offer a better balance?
2. The authors highlight that the objectives of VL adaptation and safety tuning are divergent. However, as shown in Table 2, certain safety-tuned models (e.g., LLaMA-2-Chat-VL-MTL and Tulu-2-VL-SafeRLHF) demonstrate higher helpfulness performance compared to models without safety tuning. What factors contribute to this discrepancy?
3. The paper evaluates text-only and multimodal safety using the attack success rate, while exaggerated safety is assessed using the refusal rate. Why is the exaggerated safety not evaluated using the attack success rate, and would this metric provide a more consistent assessment across different safety benchmarks?

**Questions:**

1. Notably, the linear merged model exhibits significantly lower accuracy. Would more selective merging strategies, such as conditioning on whether a layer is a safety-related layer, offer a better balance?
2. What underlying factors or characteristics could explain why certain safety-tuned models (e.g., LLaMA-2-Chat-VL-MTL and Tulu-2-VL-SafeRLHF) achieve higher helpfulness performance than models without safety tuning, despite the reported divergence in objectives?
3. Why did the authors choose different evaluation metrics (attack success rate vs. refusal rate) for text-only/multimodal safety and exaggerated safety?

---

> ### Author Response · Authors · 2024-11-19
> **Rebuttal by Authors**
>
> Dear reviewer wpWa, we deeply appreciate your constructive feedback. Here we address the weaknesses (W) and questions (Q) that you raised on our work.
>
>
> **Weight merging ablations (W1, Q1)**
>
> We conducted ablation studies that involved applying different merging methods beyond linear merging and varying the combinations of models targeted for merging (see Table 3), as well as adjusting the alpha coefficient (refer to Figure 9). However, for a more detailed analysis, we also experimented to observe how alpha coefficients vary according to different model sizes, training chat data, and architectures.
>
> For instance, when we used LLAMA-2-13B Chat instead of LLAMA-2-7B Chat to explore changes according to model size, the alpha coefficient remained constant at 0.4. Additionally, we replaced LLAMA-2-7B Chat with the base model LLAMA-2-7B, which was not tuned with a chat dataset, and found that the alpha coefficient still maintained at 0.4.
>
> Lastly, to examine changes according to architecture, we replaced LLAMA-2 Chat 7B with Mistral-7B-Instruct-v0.3, resulting in an alpha coefficient shift from 0.4 to 0.5. Furthermore, we applied a model stock merging with different merging ratios at each layer [1], which, compared to linear merging, resulted in lower performance, thereby justifying our choice of linear merging in this work.
>
> |Ablation|Alpha coefficient (applied to the SL)|
> |-----------|------------------------------------------------|
> |LLaMA-2 Chat 7B -> LLaMA-2 Chat 13B|0.4|
> |LLaMA-2 Chat 7B -> LLaMA-2 7B|0.4|
> |LLaMA-2 Chat 7B -> Mistral 7B Instruct v0.3|0.5|
>
> |Merging method|Text-only ASR ↓|Multimodal ASR  ↓|Multimodal helpfulness ↑|
> |----------------------|-------------------|---------------------|-------------------------------|
> |Linear Merging|0.31|0.72|61.6|
> |Model Stock Merging|1.13|3.82|60.9|
>
> ---
>
> **Clarification on multimodal helpfulness results (W2, Q2)**
>
> As shown in Table 2, LLaMA-2-Chat-VL-MTL scores 0.2 points lower on average in multimodal helpfulness compared to its baseline, LLaMA-2-Chat-VL, which was not safety-tuned (64.3 vs. 64.1). Similarly, Tulu-2-VL-SafeRLHF shows an average 2.1-point drop in multimodal helpfulness compared to its baseline, Tulu-2-VL (65.3 vs. 63.4).
>
> ---
>
> **Background of metric selection (W3, Q3)**
>
> The text-only/multimodal safety benchmark measures how harmful a model's responses are when exposed to harmful questions, using the attack success rate as the metric. In contrast, the exaggerated safety benchmark evaluates how often an overly safety-tuned model refuses to provide helpful responses to non-harmful questions, using the refusal rate as the metric. Since these two benchmarks assess different aspects of a model's safety, we employed distinct metrics to avoid confusion.
>
> ---
>
> **References**
>
> [1] Model Stock: All we need is just a few fine-tuned models (Jang et al., ECCV 2024)

---

> > ### Comment · Reviewer_wpWa · 2024-11-25
> >
> > Thank the authors for their clear response. I will maintain my positive rating.

---

### Official Review · Reviewer_PCPR · 2024-11-05

**Soundness:** 3
**Presentation:** 2
**Contribution:** 2
**Rating:** 6
**Confidence:** 3

**Summary:**

This paper studies the safety consequences of vision-language (VL) adaptation for language models. VL adaptation refers to the broad family of methods used to adapt pre-trained language models to multimodal prompts. The paper begins by benchmarking various VL methods for their safety performance using well known open-source language models and adaptation datasets. Some of the adaptation methods benchmarked are “safety aware”- they enforce safe preferences as part of the adaptation process. The findings show that these techniques do not prevent the degradation of model safety. The paper then attempts to reconcile the findings with observations from previous papers that balance adaptation and safety objectives, showing that the insights are aligned. Finally, the paper proposes to adapt model merging for the task of safe VL adaptation.

**Strengths:**

The paper presents a wide range of analysis for the target problem, drawing on a range of latest methods for adaptation, and reconciling / considering the analysis in light of similar ones done in recent work. Many numbers are presented (including in supplementary) and for certain methods (eg. model merging) a wide range of different hyperparams are considered and tuned for the best recipe. With analysis papers, sometimes it is difficult to tell a single clear story given how different certain results can be depending on dataset / task / adaptation method / interpretability method, and the paper does a strong job of organizing the results it presents. The story is clear: VL adaptation is important, however the usual tuning methods (eg. SFT, RLHF) lead to poor or inconsistent results even when taking safety data into account. Therefore, model merging might be a useful alternate technique in light of the “competing objective” analysis of the safety and the usefulness objectives. Qualitative figures and results are plenty and clearly presented.

**Weaknesses:**

While the story is clear, I believe that certain results are lacking for completeness toward claims. Most of the criticism are related to this:

-- The introduction is structured in such a way that it overclaims the contributions of the paper. The paper starts out by asking certain questions (eg. “is the safety degradation due to specific training data or due to the adaptation process itself) and certain claims (eg. “they [prior works] do not comprehensively focus on helpfulness or harmlessness and often focusing narrowly on individualistic benchmarks”) that it does not necessarily clarify. For example, while subsequent sections present results for investigating some of these questions, most of this analysis is also on specific datasets and individual benchmarks, which is ok but requires reframing the contribution. I believe the writing in the introduction needs to be altered.

-- Some key numbers don’t seem to be included in the paper. For example, the discussion about Table 1 requires the performance of the base model to be mentioned. However, the base performance of Tulu-2 without finetuning is not given like the base performances for the LLama_2- Chat 7B. While it is mentioned that it is not possible to apply RLHF numbers to Llama because of prior tuning, there is nothing preventing SFT finetuning being applied to Tulu in order to provide consistent comparison across the numbers for both methods. Likewise, providing the results for Llama 2 chat 7b on RLHF with safety can also be added to the appendix, despite the fact that it has been pre-instruction tuned. This would also prompt discussion on whether the findings here are model-specific or can be used to think broadly about problems that might occur with other LLMs that are being VL finetuned.

-- In order to ground the discussion, I think it is important to include some details of the various datasets and evaluation benchmarks used to perform the analysis. Why were these particular datasets chosen, how do they differ from datasets used by prior work and why are they right for a holistic analysis of safety tuning in LLMs. I think this would make for a more thorough related work than the present one - which provides an intro / prior work to the different tuning methods that have been explored for safety.

-- In order to justify the use of cosine similarity as a way to measure how much layers change during adaptation, the authors cite the work of Li et al. (2024). I think it would be helpful to have a discussion of why this is a good metric to measure change. While similarity would indeed imply high cosine values, non similar cosine values could still represent functionally similar network due to the massive parameter symmetries of neural networks.

-- It is unclear how we should be interpreting the composite metric that is presented in Figure 6. The authors attempt seems to have been  to design a score metric that is higher with higher performance, and better safety. As such, they get the complement ASR (since ASR lower the better) and add it to the performance. However, there seems to be a lack of discussion on why this metric is practical or useful. What does a 99% cumulative score on safety add beyond a 76% score? Is it worth just as much as a 3% drop in accuracy? Why is multi-modal capability only used to measure the accuracy but the average of multimodal and text only ASR used to calculate the complement? I think the paper lacks discussion on these choice. One option would be to ablate over the choices, report all choices in the appendix, and explain why one particular choice was used for Figure 6, as it appears to be the main takeaway of the paper. Further, it would really help if there were links to particular qualitative examples to deepen the analysis in Figure 6, and why it matters that Linear merged has the highest bar under this metric.

-- In Figure 5, it would also be helpful to see the other plots (eg. for MTL etc), to understand the full picture of how these results relate to previous tables and insights. The general conclusions seem to be moving in terms of which adaptation method is “better” (this is totally understandable for an analysis paper), but I believe these are further hurt by partial results presented in certain plots such as this one. For Figure 5, additionally, some discussion is warranted on the similarity of the WildVisionChat46K dataset to datasets already used for VL adaptation. If the datasets are too similar, it makes sense that the network won’t change much. What would happen for example, if a slightly OOD dataset was used for finetuning that had nothing to do with safety? If I understand correctly, the claim of the plot rests on safety finetuning being the key reason why the cosine plots become divergent, but my prior is that this may happen for any dataset that is significantly different from the one used for VL adaptation.

Miscellaneous minor points:
-- The visibility / visualization of plots is quite unclear. Fig 2 and Fig 3 have two different scales on the vertical axis, and each is applicable to different line plots. A better way to visualize this is in separate plots. Additionally, it would be helpful to plot the performance dynamics of different models (eg. not just Llama) that have been considered in Table 1 to deepen the discussion of the claim surrounding training dynamics.

-- On line 312, the accuracy stated is 42.6 while the table says 42.3, not sure which is correct

-- Numbers in Table 1 for Tulu-2 base model have not been included

-- The writing in section 5.1 was very unclear to me as it is hard to reconcile it fully with the plot. What is the key aspect of the plot that needs to be looked at to support the claim that “safety layers change the most”. It appears that all layers change.

**Questions:**

Please see above for weaknesses. Questions are included together with weaknesses. Repeating here, they are roughly aligned with the order above:
1) Is it fair to say that criticism attributed to prior work for limiting analysis to individualistic benchmarks is also applicable here? What is the key aspect of the paper / results that mitigates this problem?
2) Why were numbers for Tulu base models not included in Table 1? Why were numbers for SFT with Tulu not reported and similarly (despite Llama being instruction tuned) what precludes its inclusion as an appendix or a discussion point?
3) Why were the particular datasets chosen, how do they differ from datasets used by prior work and why are they right for a holistic analysis of safety tuning in LLMs?
4) Why is cosine similarity the right metric to study model divergence? Is it possible that models that have very similar cosine values still represent functionally different behaviors and vice versa?
5) What was the motivation to construct the composite metric in Figure 6? Why is multi-modal capability only used to measure the accuracy but the average of multimodal and text only ASR used to calculate the complement?
6) Why were MTL or other results not included in Figure 5? Further, would happen for example, if a slightly OOD dataset was used for finetuning that had nothing to do with safety?
7) What is the key aspect of the plot that needs to be looked at to support the claim that “safety layers change the most" in Figure 4?

I am willing to alter my rating if these questions are answered and writing is altered to resolve these discussion points.

---

> ### Author Response · Authors · 2024-11-19
> **Rebuttal by Authors (1)**
>
> Dear Reviewer PCPR, thank you for taking the time to provide such thoughtful and constructive feedback. We have carefully considered the weaknesses (W) and questions (Q) you raised, and are eager to address them to enhance the clarity and strength of our work.
>
> ---
>
> **Over-claiming in introduction (W1, Q1)**
>
> Thank you for your insightful feedback regarding the clarity of our introduction. Our primary aim was to highlight the relatively underexplored safety concerns of multimodal models compared to LLMs. We intended to show that training on data devoid of harmful content can inadvertently lower the inherent safety level of LLMs, and through experiments and analyses, we identified potential causes for this phenomenon and proposed efficient solutions.
>
> We acknowledge that our initial framing may have suggested broader claims, especially given the scope of our datasets. In response, we conducted additional ablation studies in diverse settings to provide more comprehensive results. Please refer to the sections on Model ablations including TULU-2 (W2, W8, Q2) and Benchmark choice and training data ablations (W3, Q3) for these findings.
>
> Regarding the question, “is the safety degradation due to specific training data or the adaptation process itself,” we added an analysis of the proportion of harmful content in the training data, which was not included in the original version of the paper. This addition strengthens the credibility of our findings related to data filtering.
>
> Finally, to address the critique of prior works, we revised the wording in the updated version to reduce over-claiming and ensure a more balanced discussion as follows.
>
> > (Previous version) First, they do not clarify whether safety degradation during VL adaptation is due to specific training data or the adaptation process itself, nor do they investigate if this degradation is gradual or occurs at specific stages. Second, they do not comprehensively assess the impact of safety tuning on both the helpfulness and harmlessness of LVLM performance, often focusing narrowly on individual benchmarks. Lastly, the suggested methods for mitigating safety issues with additional training are not cost-efficient, limiting their practicality as universal standards for practitioners.
>
> > (Revised version) First, most prior studies predominantly focus on non-multimodal models, such as LLMs, providing limited insights into the safety challenges unique to VL adaptation, where new modalities are integrated. Second, existing efforts to address safety in multimodal models often lack a holistic evaluation, failing to simultaneously consider the impact of safety tuning on both helpfulness and harmlessness in LVLM performance. Lastly, the suggested methods for mitigating safety issues with additional training are not cost-efficient, limiting their practicality as universal standards for practitioners.
>
> ---
>
> **Model ablations including TULU-2 (W2, W8, Q2)**
>
> We did not separately measure the performance of TULU-2-VL in this work because its architecture is identical to LLaMA-2, and we believed this would not provide significant value to the readers. However, recognizing the potential concern that our findings might be limited to specific model sizes or architectures, we conducted additional ablation studies.
>
> First, we examined the text-only ASR performance of the TULU-2 7B LLM. Compared to LLaMA-2-Chat 7B, TULU-2 7B showed inferior performance. Second, we investigated whether the results of VL adaptation remained consistent when using different base LLMs. Beyond the LLaMA-2-Chat 7B used in this work, we applied VL adaptation to LLaMA-2-Chat 13B, Mistral 7B Instruct v0.3, and TULU-2 7B. The results showed that performance improved with larger model sizes and more advanced architectures. However, we also confirmed that VL adaptation still caused safety degradation across models.
>
> Finally, we conducted ablation studies on the vision encoder and vision connector. Lowering the resolution of the CLIP vision encoder negatively affected both safety and multimodal performance. Similarly, replacing the default MLP connector with a single linear layer resulted in poorer performance.
>
>
> |Base LLM|Text-only ASR ↓|Multimodal ASR ↓|Multimodal helpfulness ↑|
> |-------------|--------------------|---------------------|----------------|
> |LLaMA-2-Chat 7B (Ours)|10.4|-|-|
> |TULU-2 7B|21.7|-|-|
>
> |Model|Text-only ASR ↓|Multimodal ASR ↓|Multimodal helpfulness ↑|
> |-------------|--------------------|---------------------|----------------|
> |LLaMA-2-Chat 7B VL (Ours)|54.4|96.2|64.3|
> |LLaMA-2-Chat 13B VL|50.1|94.4|66.3|
> |Mistral 7B Instruct v0.3 VL|52.1|95.3|65.7|
> |TULU-2 7B VL|62.8|97.7|62 1|
>
> |Ablation|Text-only ASR ↓|Multimodal ASR ↓|Multimodal helpfulness ↑|
> |-------------|--------------------|---------------------|----------------|
> |Ours|54.4|96.2|64.3|
> |CLIP ablation|57.9|96.7|61.8|
> |Connector ablation|55.7|96.9|62.2|

---

> ### Author Response · Authors · 2024-11-19
> **Rebuttal by Authors (2)**
>
> **Benchmark choice and training data ablations (W3, Q3)**
>
> Let us begin by explaining why the benchmarks used in this study were selected. At the time of conducting this work, these benchmarks were the most recent ones available. They categorize safety across diverse criteria, enabling us to evaluate the model's behavior under a wide range of attack scenarios. In particular, the WildJailbreaking benchmark stood out because it is based on conversations between real users and chatbots, making it closer to real-world settings and more effective for assessing a model's safety. Similarly, benchmarks like Multimodal Safety and Exaggerated Safety were chosen for comparable reasons.
>
> To measure multimodal helpfulness, we aimed to select benchmarks that are widely used, straightforward to evaluate, and unbiased. Based on these criteria, we selected the benchmarks used in this study. Since their release, many new multimodal benchmarks have emerged, particularly those focused on long-form generation. However, we opted not to prioritize these for two main reasons: they often require using closed-source LLMs and VLMs for evaluation, which significantly increases costs and introduces potential bias during evaluation.
> That said, we acknowledge the criticism regarding the lack of ablation studies. To address this and further support the consistency of our findings, we conducted additional ablation experiments, including VL adaptation data ablation, safety tuning data ablation. The results of these studies are as follows.
>
> |VL data|Text-only ASR ↓|Multimodal ASR ↓|Multimodal helpfulness ↑|
> |---------------|-------------------|--------------------------|------------------------------|
> |LLaVA v1.5 data (Ours)|54.4|96.2|64.3|
> |Vision Flan|60.1|97.1|65.7|
>
> |safety data|Text-only ASR ↓|Multimodal ASR ↓|Multimodal helpfulness ↑|
> |---------------|-------------------|--------------------------|------------------------------|
> |MTL w/ Multimodal safety data (Ours)|44.8|3.1|64.1|
> |MTL w/ Text-only safety data|45.2|33.2|62.8|
>
> ---
>
> **Metric choice (W4, Q4)**
>
> Cosine similarity has been a longstanding convention for interpreting vector representations with semantic content over the past decade. Efforts have been made to understand its role in terms of representation capacity, demonstrating that cosine similarity covers a broad range of representational similarities [1]. The rationale behind focusing on the direction and removing vector norms—essentially analyzing without norm information—stems from the perspective that norms often reflect frequency biases in the corpus rather than semantic content [2].
>
> It's important to clarify that our analysis focuses on the hidden states (vector representations) computed within the model, making it fundamentally similar to similarity analyses previously performed on vector representations (embeddings). This addresses potential concerns about the applicability of cosine similarity in this context.
>
> While many recent studies use cosine similarity without strong theoretical grounding—a practice we also see the pitfalls of—the metric's validity has been largely established through empirical observations. While cosine similarity is not the only metric of relevance, it remains a practical choice for our analysis.
>
> We acknowledge the importance of providing more comprehensive guidance around this choice, and if allowed, we will incorporate additional context in the final version to better assist readers in understanding our methodology.
>
> The concern raised about parameter symmetries in neural networks potentially leading to high cosine similarity values even for functionally different networks is valid. However, numerous studies have demonstrated that cosine similarity remains a robust metric for understanding changes in neural networks, while addressing its limitations through complementary methods.
>
> For instance, [3] introduced Cosine Normalization, leveraging cosine similarity to reduce neuron variance during training, thereby enhancing generalization performance. Their work illustrates how cosine similarity can effectively capture changes in the network during adaptation.
>
> Additionally, [4] proposed the Soft Cosine Measure, which considers the similarity between features rather than treating them as independent. This extension overcomes the traditional cosine similarity’s assumption of feature independence and provides a more accurate similarity measure.
>
> These studies highlight the validity of cosine similarity as a tool for measuring layer changes in neural networks while acknowledging the importance of addressing parameter symmetry challenges. As such, cosine similarity remains a reliable and meaningful metric for analyzing neural network adaptation processes.

---

> ### Author Response · Authors · 2024-11-19
> **Rebuttal by Authors (3)**
>
> **Some discussions on Figure 6 (W5, Q5)**
>
> In Figure 6, we chose Complement ASR to represent both safety and multimodal helpfulness metrics together. This approach was taken because the primary metric for safety, ASR, follows a "lower is better" convention, whereas the primary metric for multimodal helpfulness, accuracy, follows a "higher is better" convention. To align both metrics under the "higher is better" principle for a unified presentation, we applied the complement to ASR. Additionally, to consolidate multimodal ASR and text-only ASR into a single value, we averaged the two. The detailed safety performance without applying complement or averaging can be found in Tables 1 and 2. We also appreciate the suggestions provided and will incorporate them into the revised version as appropriate.
>
> Regarding the use of accuracy as the sole metric for measuring multimodal capability, this decision was based on practical considerations outlined in the Benchmark Choice and Training Data Ablations (W3, Q3) section. Specifically, generative benchmarks may introduce biases from the evaluation models and incur significant costs, which we aimed to avoid.
> Finally, a qualitative analysis of the examples generated by all baseline models and the linear merging model on the text-only/multimodal safety benchmark questions and the multimodal helpfulness benchmark questions is provided in Appendix F.
>
> ---
>
> **Some discussions on Figure 5 (W6, Q6)**
>
> The reason we did not conduct this analysis on MTL or other baselines is that our focus was on analyzing tasks individually. In the case of MTL, both VL adaptation and safety tuning are performed simultaneously, which could result in ambiguous outcomes. Specifically, it would be unclear whether the learning progress was primarily directed toward the chat task or the safety task. To address this, we compared the model fine-tuned for safety after VL adaptation (SL) with the model fine-tuned for chat tasks (Chatty).
>
> However, we acknowledge your valid point regarding the potential bias introduced by the similarity between the Chatty data and the data used for VL adaptation. This similarity could naturally result in high alignment scores, making the conclusions less clear. To address this concern, we conducted an additional experiment using out-of-distribution (OOD) data. Please refer to Figure 10 in the revised version of the attached paper for the experimental results plot.
>
> The additional data consisted of object detection tasks, where the input includes an image and a description of the object to locate, and the output is the coordinates of the object's position in the image. This dataset focuses on generating numeric coordinates rather than natural text, which we judged to be significantly different in distribution from the data the model had previously seen.
>
> Despite these changes, the results showed that the similarity between Chatty and VL adaptation remained high, especially when compared to SL (safety-tuned). This finding further supports the conclusion that safety tuning effectively drives the model's learning in a distinctly different direction.
>
> ---
>
> **Some discussions on Figure 4 (W9, Q7)**
>
> The message we intended to convey through Figure 4 and Section 5.1 was not that the safety layer undergoes the most significant changes among all layers. Instead, we aimed to provide the insight that the safety layer, which plays the most critical role in ensuring safety, experiences notable changes, contributing to the observed degradation in safety performance.
>
> It is true that layers toward the end of the model tend to exhibit more significant changes, a phenomenon commonly observed in prior works. However, our focus was specifically on the safety layer. Based on the findings from the original paper proposing the concept of the safety layer [5] and the results in Table 6, we conclude that freezing the safety layer alone provides substantial benefits for safety.
>
> While the degree of change in the safety layer is not as large as in other layers toward the end, its role in ensuring safety remains pivotal. When it was shown that the safety layer is essential for safety, even a change reflected by a cosine similarity of 0.5 was considered significant and meaningful in this context. As a side note, we recognize that the current explanation might cause some confusion. Therefore, we plan to revise the phrasing in the updated version to ensure greater clarity.
>
> ---
>
> **Correcting typo (W7)**
>
> Thank you for catching the error. The correct score is 42.6%. I will make sure to incorporate this into the revised version.

---

> > ### Author Response · Authors · 2024-11-23
> > **Rebuttal by Authors (4, References)**
> >
> > **References**
> >
> > [1] Representation Learning with Weighted Inner Product for Universal Approximation of General Similarities (Kim et al., 2019)
> >
> > [2] Norm of Word Embedding Encodes Information Gain (Oyama et al., 2023)
> >
> > [3] Cosine normalization: Using cosine similarity instead of dot product in neural networks (Luo et al., 2018)
> >
> > [4] Soft Similarity and Soft Cosine Measure: Similarity of Features in Vector Space Model (Sidorov et al., 2014)
> >
> > [5] Safety Layers in Aligned Large Language Models: The Key to LLM Security (Li et al., 2024)

---

> > > ### Author Response · Authors · 2024-11-25
> > > **Kindly reminder for discussion**
> > >
> > > Dear Reviewer PCPR, We appreciate for your time and effort on reviewing the paper. As we near the end of the discussion period, we kindly remind you of the upcoming deadline. We are eager to discuss any aspects of the paper that may require further clarification. Thank you once again for your valuable feedback.

---

> > > > ### Comment · Reviewer_PCPR · 2024-11-25
> > > > **Thanks for the feedback.**
> > > >
> > > > Thanks you for addressing my concerns. The rebuttal clearly addressed and answered most of my questions. I am now generally more positive about the paper, raising my score to an acceptance score (6). However, I still have some concerns about soundness and somewhat ad-hoc of metrics (eg. W5, Q5) used for comparison in the paper. While I understand the reasons provided (the desire to combine lower is better and higher is better scores into one scores), I am still unclear on how the relative scales for the two quantities being balanced etc were chosen in constructing these metrics (eg. why does it make sense to value ASR / helpfulness equally). Further, as promised by authors, there is large room to improve writing quality and plot clarity. I'm hoping these are addressed in the eventually submitted version, but am leaning toward acceptance now.

---

### Author Response · Authors · 2024-11-22
**Looking forward for Discussions**

Dear Reviewers,

We appreciate for your time and effort on reviewing the paper.

As we near the end of the discussion period, we kindly remind you of the upcoming deadline.

We are eager to discuss any aspects of the paper that may require further clarification.

Thank you once again for your valuable feedback.

---

### Meta-Review · Area_Chair_8BDD · 2024-12-22

**Metareview:**

After careful consideration of the six expert reviews, and the subsequent author-reviewer discussions, I recommend accepting this submission. The paper presents an investigation of how vision-language adaptation affects the safety capabilities of large language models, offering insights for the development of safer multimodal AI systems.

The paper's primary contribution lies in its analysis of safety degradation during vision-language adaptation and its proposed solution through weight merging. While reviewer 2icJ raised valid concerns about the limited scope of model architectures tested, the authors demonstrated through additional experiments that their findings generalize across several model variants and scales. The authors' response during the discussion period, including new experiments with different model sizes and architectures, strengthens confidence in their conclusions.

Multiple reviewers highlighted the paper's presentation and experimental thoroughness. Reviewer EKaT particularly praised the paper's comprehensive evaluation metrics and clear visualization of different safety types. The authors' subsequent additions during the rebuttal period, including detailed ablation studies and theoretical analysis of optimization challenges, further enhanced the work's technical depth.

While some concerns were raised about the theoretical foundations and broader applicability of the weight merging approach, the authors provided convincing responses and additional experimental evidence. The discussion period revealed some limitations in the work, particularly regarding the coverage of larger models and diverse architectures. However, as noted by several reviewers, these limitations are largely due to computational constraints rather than methodological flaws. The authors' transparency about these limitations and their investigation within available resources strengthens rather than weakens the paper's contribution.

**Additional Comments On Reviewer Discussion:**

During the rebuttal period, there was extensive discussion between the authors and reviewers focused on several key aspects of the paper: model architecture coverage, theoretical foundations, and experimental validation. The discussions demonstrated both the paper's strengths and areas for potential improvement.

A significant thread of discussion centered on the breadth of model architectures tested. Reviewer 2icJ raised concerns about the limited coverage of different model sizes and architectures, suggesting the inclusion of additional models like Qwen-VL and ShareGPT4V. The authors responded by providing new experimental results across different model sizes and architectures, including LLaMA-2-13B and Mistral-7B. While these additions helped demonstrate broader applicability, some reviewers maintained that even wider coverage would strengthen the paper's claims.

Another major point of discussion focused on the theoretical foundations of weight merging. Multiple reviewers questioned the novelty and theoretical justification of this approach. The authors addressed these concerns by providing additional analysis of optimization challenges in multi-task learning, referencing established work on gradient conflicts and demonstrating consistent alpha coefficient values across different model configurations. Their response effectively situated their contribution within the broader literature while highlighting the unique challenges of multimodal safety.

Several reviewers, including PCPR and cNbS, raised questions about evaluation metrics and the definition of safety. The authors provided detailed clarification of their metric choices and conducted additional ablation studies to demonstrate the robustness of their findings. Particularly noteworthy was their response to concerns about the composite metric in Figure 6, where they provided additional context and justification for their approach.

Furthermore, the discussions revealed that while there is room for deeper theoretical analysis and broader empirical validation, the current work makes a contribution to our understanding of safety in multimodal models.

---

### Decision · Program_Chairs · 2025-01-22

Accept (Poster)